# A Window of Vulnerability: Chronic Environmental Stress Does Not Impair Reproduction in the Swordfish *Xiphias gladius*

**DOI:** 10.3390/ani13020269

**Published:** 2023-01-12

**Authors:** Giorgia Gioacchini, Sara Filippi, Rossella Debernardis, Luca Marisaldi, Riccardo Aiese Cigliano, Oliana Carnevali

**Affiliations:** 1Department of Life and Environmental Sciences, Polytechnic University of Marche, 60131 Ancona, Italy; 2Sequentia Biotech SL, 08005 Barcellona, Spain

**Keywords:** oogenesis, health, puberty, RNA-seq, melanomacrophage

## Abstract

**Simple Summary:**

The Mediterranean swordfish (*Xiphias gladius*) stock was affected in last decades from overfish. The evaluation of the cross-talk among metabolism, stress response, immune system and reproduction in immature and mature females could be of great importance. For these reasons, in the present study, by transcriptomic and histological analysis, a deeper insight into the cross-talk among reproduction, metabolism and response to environmental cues across different stages of sexual maturation was provided by the livers of female swordfish. We show that mature females are able to properly reproduce, since they invest a lot of energy in reproduction; however, their detoxification capacity and immune system are compromised as evidenced by transcriptomic and histological approaches. These results suggest that during the reproductive season, mature females may be more susceptible to environmental stress and pollutants than immature ones.

**Abstract:**

*Xiphias gladius* is an important fishing resource. The Mediterranean stock is affected by overfishing and is declining. In this light, the aim of this study was to evaluate the cross-talk among metabolism, stress response, immune system and reproduction in immature and mature females, coupling histological and transcriptomic approaches. The transcriptome of livers from 3 immature and 3 mature females was analyzed using the Artificial Intelligence RNA-Seq. For the histological analysis, ovary and liver samples were collected from 50 specimens caught during the reproductive season in the Mediterranean Sea. A total of 750 genes were differentially expressed between the livers. The gene ontologtabey analysis showed 91 upregulated and 161 downregulated biological process GO terms. Instead, the KEGG enrichment analysis revealed 15 enriched pathways. Furthermore, the binding occurring between estrogen receptors and aryl hydrocarbon receptor nuclear translocator, upregulated in mature females, could be liable for the inhibition of detoxification pathway. Indeed, at the histological level, mature females showed a higher density and number of melanomacrophage centers, biomarkers of stress. The present findings reveal the cross-talk among response to environmental stressors, metabolism and reproduction, highlighting that mature females invest a lot of energy in reproduction instead of immune response and detoxification.

## 1. Introduction

The swordfish (*Xiphias gladius*) is a cosmopolitan, highly migratory teleost species and an important fishing resource. The Mediterranean stock has been affected by overfishing and has declined since the 1980s [1]. A recovery plan was established by the International Commission for the Conservation of the Atlantic Tunas (ICCAT), including measures such as fishing fleet capacity limitations, closed fishing season and a minimum size [1]. Recently, more attempts to better understand the reproductive biology of this fish (i.e., spawning area and period) and sexual maturity were carried out with promising results [2,3,4,5,6]. In this context, deepening the knowledge about metabolic requirements during reproduction, growth pattern and responses to external stimuli are central points. Thus, a deep knowledge at the molecular level of the genes underlying these processes is necessary to obtain a more complete picture of the reproductive potentiality of this species. Recently, RNA-seq was widely applied in several studies for the conservation and preservation of the fisheries resource, investigating genes involved in adaptation to environmental changes [7,8]. Notably, swordfish ovarian and liver transcriptome analysis was performed [4,9]. In particular, the transcriptome of ovaries from mature and immature females, caught in the breeding season, were investigated in order to identify and characterize the molecular network involved in sexual maturity and in the circadian rhythm. Despite these papers focusing attention on the genes involved in sexual maturation and reproduction such as estrogen receptor α (esrα) and three forms of vitellogenin, little attention was paid to genes involved in metabolic processes and responses to environmental stimuli. How the limited internal resources of any organism flow among reproduction, growth and response to environmental stress is a key determinant of the success of survival of the species [10,11]. In wild populations, this energetic balance is challenged by chronic and acute environmental stress such as chemical pollution [12], habitat modification [13] and climate change [14]. In teleost, in fact, an important target organ of such environmental stressors is the liver, and its health status is used for biomonitoring purposes [15]. The liver is responsible for several processes, such as immune response [16,17], detoxification [18], metabolism [19,20] and reproduction [21,22]. In the liver, the lipid metabolism is investigated to identify the genes and pathways involved in fatty acid synthesis, lipid transport and the oxidation process [23]. Numerous studies were conducted to understand the lipid metabolism in fish species exposed to different diets, toxic compounds or nanoplastics [24,25,26]. In addition, in fish, the hepatic lipid metabolism (synthesis and mobilization) plays an important role during oogenesis, (i.e., yolk formation or oocyte lipidation) and in further larval development [27,28]. Furthermore, the histological analysis of the liver can reveal the health status of several fishes by evaluating the occurrence of melanomacrophages (MM), an important environmental biomarker [29,30,31,32,33], which are involved in detoxification, immune response and destruction/recycling of various exogenous and endogenous materials, such as erythrocytes (ferric ion), pathogens [34,35] and nanoparticles [36,37]. An increase in aggregate of melanomacrophages (MMC) or single melanomacrophage (MMs) cell density has been recorded along with increasing expression levels of several genes involved in detoxification and reproduction in fishes after exposure to different stressors [34,38,39,40].

The coupling of RNA-seq and histological approaches has been widely used to investigate the effect of pollutants at the molecular and physiological level [41,42,43].

In fact, changes in transcripts abundance revealed a modification of molecular signaling. However, the transcriptome changes should be linked to a physiological response because they could not reflect the real RNA translation [43].

In the present study, by transcriptomic analysis, a deeper insight into the cross-talk between reproduction, metabolism and response to environmental cues across different stages of sexual maturation was provided by the livers of female swordfish. In addition, by histological assessment of MMC and MMs density/frequency, size and lipid content, we successfully identified a variation related to sexual maturity and fish size. Taken together, these results reveal the molecular relationship between metabolism, response to environmental stressors and reproduction, highlighting that mature females invest most of their energy in reproduction instead of detoxification and immune responses.

## 2. Materials and Methods

### 2.1. Sample Collection

Liver samples were collected from swordfish females already caught and analyzed in previous studies using different approaches and for different purposes [3,4,9] (Appendix A). Briefly, the animals were caught by commercial long-liners in the central and western Mediterranean Sea. The lower jaw to fork length (LJFL) (min = 97 cm; max = 190 cm) and total weight (TW) (min = 8 kg; max = 90 kg) were recorded for each specimen. Reproductive status of females was previously determined by histological analysis [3,4,9], and 22 immature and 28 mature females were included in this study. The samples were collected following the guidelines of the International Commission for the Conservation of Atlantic Tuna (ICCAT). Samples of livers (~2 cm^3^) were fixed in a formaldehyde-glutaraldehyde solution (NaH_2_PO_4_·H_2_O + NaOH + formaldehyde 36.5% + glutaraldehyde 25% + H_2_O) and stored at 4 °C until histological analyses.

### 2.2. Transcriptomic Analysis

In order to focus on the molecular cross-talk among reproduction, metabolism and response to environmental cues between immature and mature females, we leveraged the knowledge contained in the transcriptome recently assembled and described by Gioacchini and collaborators (Appendix A) [4]. The experimental dataset (Illumina paired-end 150 bp reads) of livers from 3 immature and 3 mature females was downloaded from SwordfishOmics (http://www.swordfishomics.com, accessed on 29 May 2020). Reads mapping, using to reference the swordfish genome, and differential gene expression analysis (FDR cut-off < 0.05) were performed using the A.I.R. (Artificial Intelligence RNA-Seq) software from Sequentia Biotech (https://transcriptomics.sequentiabiotech.com/, accessed on 29 May 2020), which applies empirical Bayes estimation and exact tests based on a negative binomial model (edgeR). The immature female transcriptome was used as reference group. Gene ontology analysis was performed in the A.I.R. environment. The gene ontology enrichment analysis (GOEA), based on differentially expressed genes, was performed using the clusterProfiler package [44] in the RStudio environment, and the *p*-values were adjusted with the Benjamini-Hochberg method [45]. Then, the enriched GO terms were investigated based on the research questions, by analyzing the enrichment score and the number of genes for each GO term. In addition, the KEGG enrichment analysis and BRITE functional hierarchies analysis (A-B categories) were carried out with the clusterProfiler package [44] in the RStudio environment. The *p*-values were adjusted with the Benjamini-Hochberg method [45]. Pathways enriched by differentially expressed genes were investigated based on the topic of interest.

### 2.3. Experimental Validation

Validation of five genes (*elovl6, fabp1, igf-1, ers1, sbrepb1*) was performed by means of qPCR. From samples selected for transcriptomics analysis, a total amount of 1 µg of RNA was used for cDNA synthesis, employing the iScript cDNA Synthesis Kit (Bio-Rad, Hercules, CA, USA). PCRs were performed with the SYBR green method in a CFX96 Real-Time PCR system (Bio-Rad) following Gioacchini and coworkers [46]. Acidic ribosomal phosphoprotein P0 (*arp*) and ribosomal protein L7 (*rpl7*) were used as internal standards in order to standardize the results by eliminating variation in mRNA and cDNA quantity and quality. No amplification products were observed in negative controls, and no primer-dimer formations were observed in the control templates as indicated by the melting curve analysis. The data obtained were analyzed using the CFX Manager Software version 3.1 (Bio-Rad), including GeneEx Macro Conversion and GeneEx Macro files and results represented by bar-plots along with the standard error. Statistical significance was attained using a *t*-test. Specific primer pairs for target genes (*elovl6:* Fw- ATATGGCCTTGTGGCTTCC, Rv- GCCATTCTGGTGCTCCTTCT; *fabp1:* Fw- GCATGAGGGGCGGATAGGAA, Rv- AAGGTCCCAGTTACCTCCACGATA; *igf-1:* Fw- TGTAGCCACACCCTCTCACT, Rv- GGGCCATAGCCTGTTGGTT; *ers1:* Fw- GACAAACGACGAACTGGCAC, Rv- CTCCCATCCTGAAGGAGCAC; *sbrepb1:* Fw- CCTGTCTAAAGGCCCTCGGT, Rv- TTAGCAGAGACCACAACGCA; *arp*: Fw- ACAGCCCAGTCTTTCCACAG; Rv- TTTAAGGTCCGGGCAACCTG; *rpl7*: Fw- GTACTGCTCGCAAAGTGGGA, Rv- GACTTTGGGGCTGACACCAT) were designed with Primer-Blast.

### 2.4. Histological Analysis

Liver samples were serially dehydrated in graded ethanol, cleared in xylene and embedded in paraffin. Sections of 4 μm were cut with a microtome (model RM2125 RTS; Leica Biosystems, Wetzlar, Germany), stained with Mayer’s haematoxylin/eosin and examined under a microscope (Axio Imager 2; Zeiss, Oberkochen, Germany). Quantification of MMCs and MMs was performed in 5 sections taken with a 20x objective (digital field area = 149,738 μm^2^), while the quantification of lipids was performed in 5 sections with a 40x objective (digital field area = 39,533 μm^2^) [47]. The separation between each section was 40 μm. The density of MMCs and their number per mm^2^ of hepatic parenchyma as well as the density of lipids were measured using Fiji [48]. The density of MMCs, MMs and the density of lipids were expressed as μm^2^/mm^2^. Data were first checked for normality with the Shapiro test, and a Pearson’s correlation test was performed among the density and number of MMs and MMCs, the density of lipids and TW and LJFL (kg and cm, respectively) in the R-studio environment using the cor.test command. The *t*-test was performed between the immature and mature groups for each variable using GraphPad Prism 6 version 6.00 for Windows (GraphPad Software; La Jolla, CA, USA)).

## 3. Results

### 3.1. Transcriptomic Analysis

In order to investigate the dynamics occurring in swordfish among immune system, metabolism, reproduction and stress response during the breeding season, the transcriptomes between mature and immature livers assembled by Gioacchini and collaborators were analyzed using a different protocol of statistical analysis and the reference genome [4].

#### 3.1.1. DEGs

To deepen the understanding of molecular cross-talk among reproduction, metabolism, immune system and stress response, the focus was placed on DEGs. In particular, twenty-one upregulated and twenty-one downregulated genes were identified for each process of interest (Figure 1). Five upregulated genes, including aryl hydrocarbon receptor nuclear translocator-like 2 (*arnt2*), and four downregulated genes, including cytochrome P450 family 1 subfamily A polypeptide 1 (*cyp1a1*), hepatocyte nuclear factor 4 alpha (*hnf4a*) and glutathione peroxidase (*gpx*), are involved in stress response. Five upregulated genes, including B-cell receptor CD22 B-lymphocyte cell adhesion molecule (*cd22*) and immunoglobulin superfamily member 8 (*igsf8*), and eight downregulated genes, including complement component 8, beta polypeptide (*c8b*), complement component pro-C3 (*c3*) and complement factor H (*cfh*), are involved in the immune system. Eight upregulated genes, including fatty acid-binding protein liver-type (f*abp1*) phospholipid-transporting ATPase (*drs2*), and eight downregulated genes, including elongation of very-long-chain fatty acids protein 6 (*elovl6*) and preproinsulin-growth factor I (*igf1*), are involved in metabolism. Three upregulated genes, including estrogen receptor alpha short form (*esr1*), and one downregulated gene, estrogen receptor beta (*esr2*), are involved in reproduction. Five of the DEGs identified were quantified by qPCR, and the differences were validated (Figure 2).

#### 3.1.2. Gene Ontology

Using the reference genome of swordfish and the empirical Bayes estimation and exact tests based on a negative binomial model, a total of 750 differentially expressed genes (DEGs) were identified between immature and mature females: 355 upregulated and 395 downregulated. The GOEA (gene ontology enrichment analysis) identified fourteen enriched biological processes, including lipid transport (GO:0006869), response to estradiol (GO:0032355), response to bacterium (GO:0009617), egg coat formation (GO:0035803) and response to polycyclic arene (GO:1903165) (Figure 3).

The gene ontology analysis showed 91 upregulated and 161 downregulated biological process GO terms. Twenty-eight upregulated and twenty-nine downregulated biological process GO terms were examined based on their involvement in the immune system, metabolism, reproduction and stress response. The livers from mature females showed upregulated GO terms such as response to polycyclic arene (GO:1903165) involved in stress response, response to bacterium (GO:0009617) involved in immune system, long-chain fatty acid transport (GO:0015909) involved in metabolism and response to estradiol (GO:0032355) involved in reproduction, and downregulated GO terms such as xenobiotic metabolic process (GO:0006805) involved in stress response, complement activation (GO:0006956) involved in immune system, lipid metabolic process (GO:0006629) and cellular response to estrogen stimulus (GO:0071391) (Figure 4).

#### 3.1.3. KEGG Enrichment Analysis

In order to investigate the dynamics occurring in the swordfish liver, 203 upregulated and 105 downregulated genes with KEGG annotation were mapped onto the Kyoto Encyclopedia of Genes and Genomes database. KEGG BRITE functional hierarchies analysis was performed at A (macrocategories) and B (subcategories) levels using the number of DEGs as the variable (Figure 5 and Figure 6). The most representative KEGG macrocategories were related to metabolism (DEGs > 100), response to environmental information processing (DEGs > 60), cellular processes (DEGs > 15) and human diseases (DEGs > 10). The most representative subcategories were related to lipid metabolism (DEGs > 30), transport and catabolism (DEGs > 10) and immune system (DEGs > 5). The KEGG enrichment analysis revealed a total of 15 enriched pathways, 6 upregulated and 9 downregulated. Among them, in mature females, steroid biosynthesis (*map00100*) and fatty acids degradation were upregulated, while metabolism of xenobiotics by cytochrome P450 (*map00980*), complement and coagulation cascades (*map04610*) biosynthesis of unsaturated fatty acids were downregulated (Figure 7 and Figure 8).

### 3.2. Histological Analysis

Thanks to a histological classification according to Marisaldi et al. [3], 22 immature and 28 mature females were identified. Among mature ones, 17 mature females showed a developing ovary, while 11 mature females showed a spawning capable ovary.

The histological appearance of female livers is shown in Figure 9. Liver parenchyma appeared homogeneous with polygonal-shaped hepatocytes having spherical nuclei; the lipids appeared as white dots and were distributed homogenously. MMs were located in all parenchyma and near blood vessels and sinusoids and appeared as dark brown dots. MMCs were located attached to the blood vessels or bile ducts and appeared with a color ranging from dark brown to light brown; structured and unstructured morphologies and highly varied sizes were evident.

#### 3.2.1. MMs and MMCs Related to Fish Size

The density and number of MMCs and MMs were correlated with fish size (LJFL and TW) (Table 1). A significant positive correlation was observed between fish weight and both density and number of MMCs (*p*-value < 0.05, Pearson’s correlation = 0.64 and 0.81, dMMCs and MMCs/mm^2^, respectively). A similar result was observed between fish length and both density and number of MMCs (*p*-value < 0.05, Pearson’s correlation = 0.62 and 0.83, dMMCs and MMCs/mm^2^, respectively). In contrast, a significant negative correlation was observed between fish weight and dMMs (*p*-value < 0.05, Pearson’s correlation = −0.53), whereas no significant correlation was observed between fish weight and MMs/mm^2^ (*p*-value > 0.05, Pearson’s correlation = −0.43). A significant negative correlation was observed between fish length and both density and number of MMs (*p*-value < 0.05, Pearson’s correlation = −0.6283032 and −0.5467768, dMMs and MMs/mm^2^, respectively).

#### 3.2.2. MMs and MMCs Related to Fish Sexual Maturity

Differences in MMCs and MMs density and number were also investigated in relation to sexual maturity as histologically established.

The density of MMCs showed a significant increase in mature females compared to immature females (*p*-value < 0.05), while the density of MMs showed a strong significant decrease in mature females with respect to immature females (*p*-value < 0.001). The number of MMCs per mm^2^ liver parenchyma showed a strong significant increase in mature females compared to immature females (*p*-value < 0.001), while the number of MMs per mm^2^ showed a strong decrease in mature females compared to immature females (*p*-value < 0.001) (Figure 10).

#### 3.2.3. Lipids Related to Fish Size and Sexual Maturity

The density of lipids was correlated to fish size (LJFL and TW) (Table 2).

A negative but not statistically significant correlation was observed between fish weight and lipids (*p*-value > 0.05, Pearson’s correlation = −0.40). Instead, a significant negative correlation was observed between fish length and lipids (*p*-value > 0.05, Pearson’s correlation = −0.53) (Table 2). Moreover, the density of lipids was related to sexual maturity. The density of lipids showed a significant decrease in mature females compared to immature females (*p*-value < 0.05) (Figure 11).

## 4. Discussion

The high-throughput sequencing technology has become a good method to identify and characterize the cross-talk and interactions among many biological processes [49]. Furthermore, the coupling of RNA-seq with histological analysis can elucidate the dynamics occurring during the reproductive season and the sexual maturation of marine fish species and support a science-based decision-making process in the context of fishery management [41,42,43]. In the present study, we highlighted the differences in metabolism, reproduction, immune system, and stress response between immature and mature swordfish females during the breeding season. Focusing on metabolism, a higher density of lipids was found in the liver of immature females than mature females. A similar result was described by Zudaire et al. [50] in yellowfin tuna (*Thunnus albacares*). These results could be attributed to a different use of energy between mature and immature females. In mature females, during the reproductive season, lipids were used for the hepatic synthesis of vitellogenin and neutral lipids which will be uptaken by the oocytes during vitellogenesis [50,51]. These results were also confirmed by transcriptomic analysis. The GOEA evidenced a prevalence of upregulated GO terms related to lipid transport and mobilization in mature females. On the contrary, the GO terms involved in the biosynthesis of metabolites, including fatty acids, are downregulated, and this result could explain the lower lipid density found in the liver of mature females. In immature females, the lipids were used for somatic growth, as confirmed by the overexpression of genes such as *igf1*, *igfbp2*, *ghe*, and *Irs1*. The mature females, investing energy in reproduction, could not have enough energy to invest in immune system and to withstand the stress. Effectively, the mature females showed a downregulation of genes related to the immune system (*il4r*, *c8b*, *c9*) and detoxification (*cyp1a1*, *cyp1b1*), and an upregulation of genes related to the response to polycyclic aromatic hydrocarbons (*arntl2*) and response to cadmium (*hspa1s*). In addition, Casanova-Nakayama et al.’s [52] study of rainbow trout (*Oncorhynchus mykiss*) indicated that the estrogens, present at high levels in mature females during the reproductive period [53], could inhibit the immune system, inducing immunosuppressive effects and therefore exposing the animal to infections. In addition, several studies showed an inhibitory action of the estrogens towards interleukins (il) [54,55]. *Xiphias gladius* is an apical predator and is considered a “reservoir” of pollutants. The presence of high concentrations of persistent organic pollutants (Pops) and trace metals (cadmium, mercury, arsenic) was previously confirmed in the liver of Mediterranean swordfish by other studies [56,57]. Many of these compounds interact with the aryl hydrocarbon receptor (*ahr*), which binds a nuclear translocator (*arnt*). This complex binding specific DNA sequences, the xenobiotic response elements (*xre*), activates the transcription of genes encoding for enzymes (*cyp1a1*, *cyp1a2*, *cyp1b1*) involved in the detoxification of xenobiotics and drugs [58]. Our results showed a downregulation of *cyp1a1* and *cyp1b1*, while *arnt2* was upregulated. These results could be explained by the fact that in the presence of high levels of estradiol, arnt2 bind estrogen receptors (ERs) induce several mechanisms, including both up- and downregulation of ERs transcription and degradation of ERs (proteosome) [59,60,61,62]. In this light, the competition between AhR and ERs for the same cofactor (arnt2) could inhibit the cytochrome p450 signaling pathway [63] and modulate the ERs gene expression, suggesting that the detoxification capacity of *cyps* is reduced in mature females characterized by high levels of estrogen due to reproductive regulation. The transcriptomic results are in agreement with the histological ones. MMCs are involved in destruction/recycling of various exogenous and endogenous materials [35,40,64,65]. In our study, the mature females showed a significantly higher density of MMCs than immature females. This result suggests that the increase of MMCs could be due to the downregulation of genes involved in xenobiotics metabolism, such as *cyp1a1* and *cyp1b1*, and therefore, melanomacrophages are the only mechanisms that undertake the detoxification function. In this study, the increase in density and number of MMCs were found to be positively and significantly correlated to the fish size (LJFL and TW). These results are in agreement with the positive correlation between MMCs and the age previously described in several teleost species [47,66]. Notably, a recent study on the European anchovy demonstrates that long-term exposure to contaminated waters increases the presence and density of MMCs and MMs. Indeed, these results suggest that mature females, which are older than immature ones, are exposed for a long time to stressors or pollutants. Furthermore, the immature females showed a higher density of MMs compared to mature females. This result is in accordance with transcriptomic ones, which revealed that the immature females respond to pollutants by a more reactive immune and detoxification system. In addition, immature females are not in reproduction, suggesting that the pathway involved in detoxification is not inhibited by estrogens, as confirmed by the transcriptomic analysis. Moreover, the immature females are not exposed to stressors or contamination for long periods: in fact, the density and number of MMs is significantly correlated in a negative way to fish size, suggesting that immature females are exposed to stress for a while, then to acute stress. In *Poecilia reticulata*, it was evidenced that the number and density of MMs increased until 7 months and decreased after this age [47], while the density and number of MMCs increased after 7 months. Our study indicates that immature and mature females show differences during the reproductive season, both in the number and density of melanomacrophages, in hepatic lipid density and in the expression of genes. In addition, both swordfish females show no optimal health status, but mature females seem to have more difficulty responding to chemical and chronic stress during the reproductive season.

## 5. Conclusions

In conclusion, the results obtained in this work reveal that during the reproductive season, mature females invest most of their energy in reproduction instead of detoxification and immune response. For this reason, during the reproductive season, mature females may be more susceptible to environmental stress and pollutants, also due to the inhibition of detoxification and the immune system. In this light, further studies on mature and immature females during the nonreproductive season could add new information on the health status of Mediterranean swordfish, to assess whether the immune-deficient situation of mature females persists or if it is more linked to the reproductive season.

## Figures and Tables

**Figure 1 animals-13-00269-f001:**
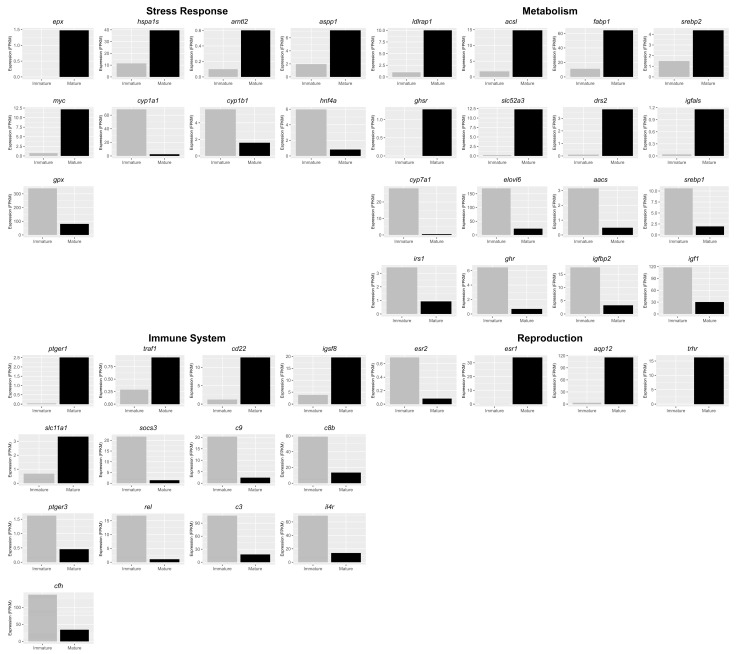
DEGs analysis between livers of immature (*n* = 3) and mature (*n* = 3) females. Bar plot shows gene expression levels (FPKM) of genes involved in metabolism, immune system, reproduction and stress response. *y*-axis indicates the FPKM level.

**Figure 2 animals-13-00269-f002:**
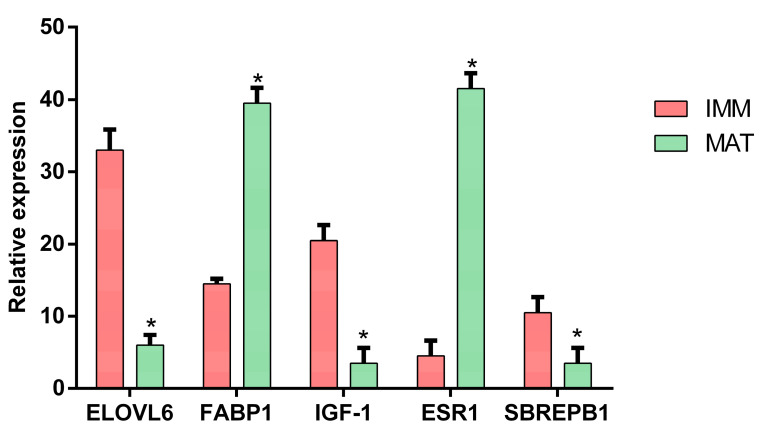
qPCR validation. Bar plot shows gene expression levels of five genes, measured by qPCR between livers of immature (*n* = 3) and mature (*n* = 3) females. Significance between mature and immature for each gene is shown by an asterisk, and standard error is reported with error bars.

**Figure 3 animals-13-00269-f003:**
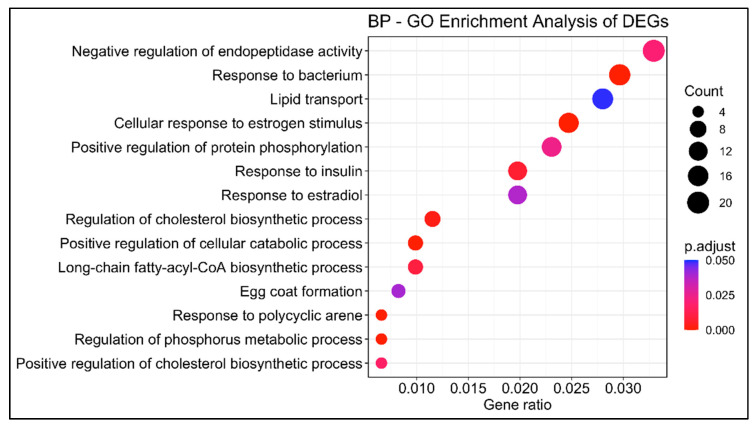
Gene ontology enrichment analysis of differentially expressed genes (DEGs) between livers of immature (*n* = 3) and mature (*n* = 3) females. *y*-axis indicates the GO term description; *x*-axis indicates the gene ratio. The size of the dot is based on gene count, and the color of the dot shows the GO term’s enrichment significance (*p*-adjust < 0.05).

**Figure 4 animals-13-00269-f004:**
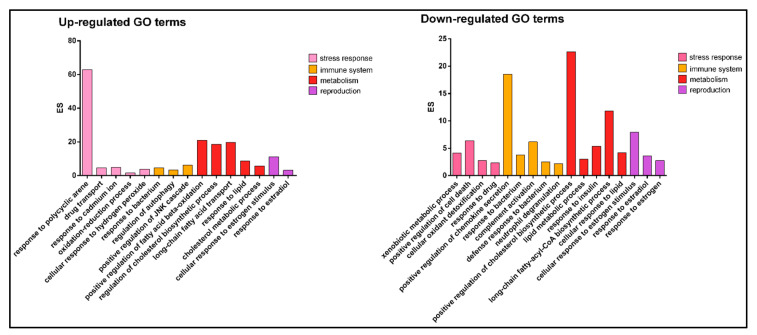
The up- and downregulated GO terms examined based on their role in immune system, metabolism, reproduction and stress response. *y*-axis indicates the enrichment score (ES); *x*-axis indicates the GO term description. The colors indicate the processes examined.

**Figure 5 animals-13-00269-f005:**
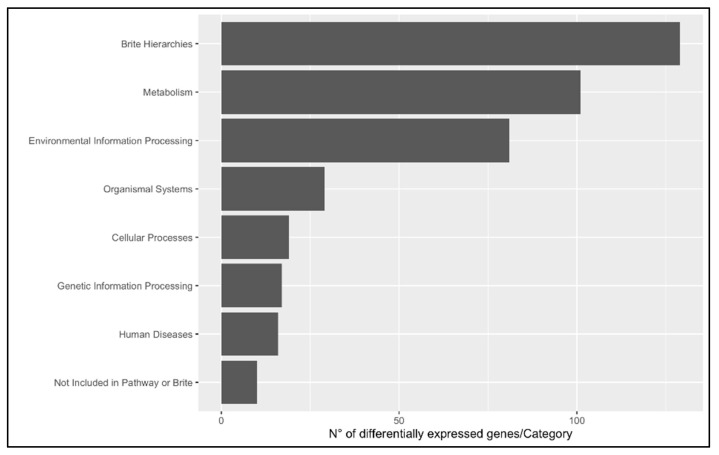
KEGG BRITE functional hierarchies analysis category A (macro category). The bar plot shows in *x*-axis the number of DEGs per macrocategory between livers of immature (*n* = 3) and mature (*n* = 3) females; *y*-axis indicates the pathway description.

**Figure 6 animals-13-00269-f006:**
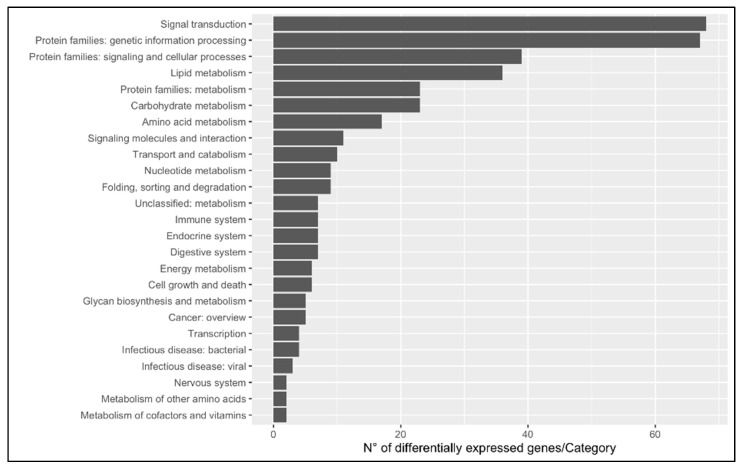
KEGG BRITE functional hierarchies analysis category B (intermediate category). The bar plot shows on *x*-axis the number of DEGs per intermediate category between livers of immature (*n* = 3) and mature (*n* = 3) females; *y*-axis indicates the pathway description.

**Figure 7 animals-13-00269-f007:**
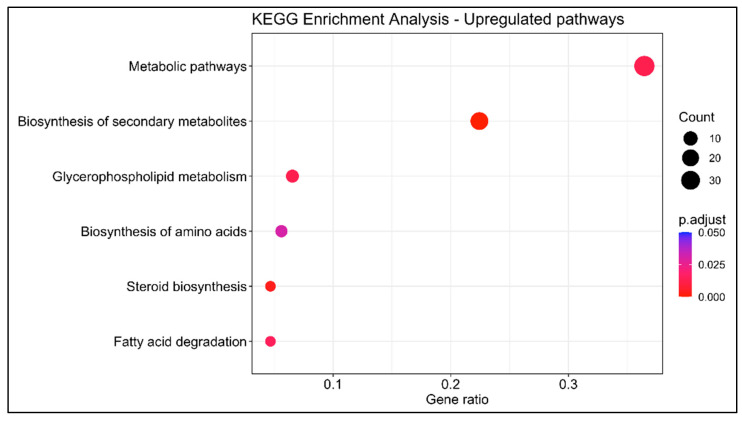
KEGG enrichment analysis of differentially upregulated genes between livers of immature (*n* = 3) and mature (*n* = 3) females. *y*-axis indicates the pathway description; *x*-axis indicates the gene ratio. The size of the dot is based on gene count, and the color of the dot shows the GO term’s enrichment significance (*p*-adjust < 0.05).

**Figure 8 animals-13-00269-f008:**
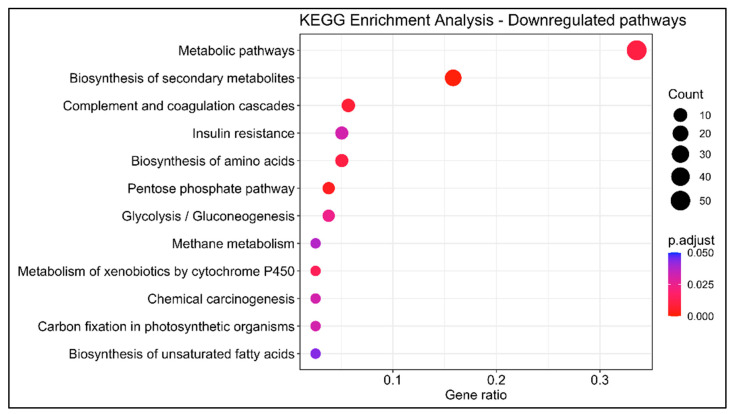
KEGG enrichment analysis of differentially downregulated genes between livers of immature (*n* = 3) and mature (*n* = 3) females. *y*-axis indicates the pathway description; *x*-axis indicates the gene ratio. The size of the dot is based on gene count, and the color of the dot shows the GO term’s enrichment significance (*p*-adjust < 0.05).

**Figure 9 animals-13-00269-f009:**
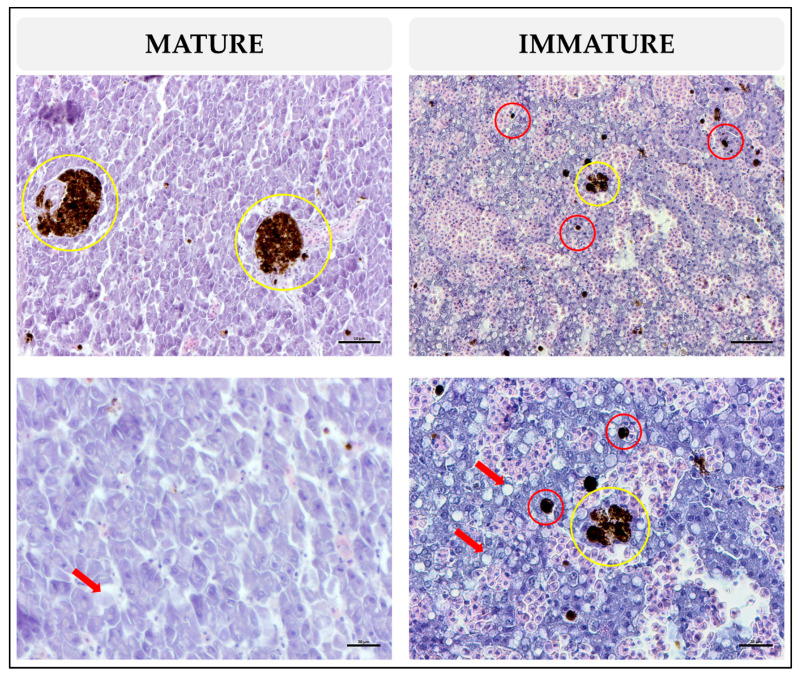
Photomicrograph of the immature and mature liver of *Xiphias gladius*. Hematoxylin and eosin (H&E) staining of a section of the liver shows lipid components (red arrow); melanomacrophage center presence (yellow circle); single melanomacrophages (red circles).

**Figure 10 animals-13-00269-f010:**
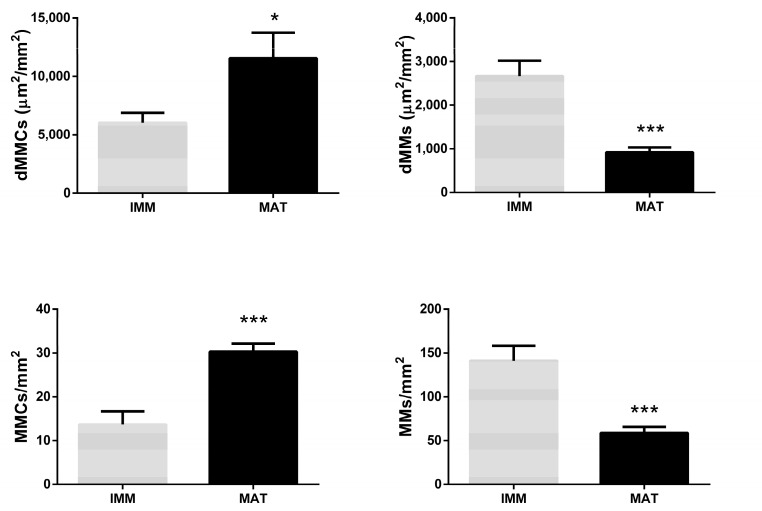
Differences in density and number per mm^2^ of single (MMs) melanomacrophages and centers (MMCs) between livers of immature (IMM; *n* = 22) and mature (MAT; *n* = 28) females. Asterisk (*) indicates significant statistical difference (* = *p*-value < 0.05, *** = *p*-value < 0.001).

**Figure 11 animals-13-00269-f011:**
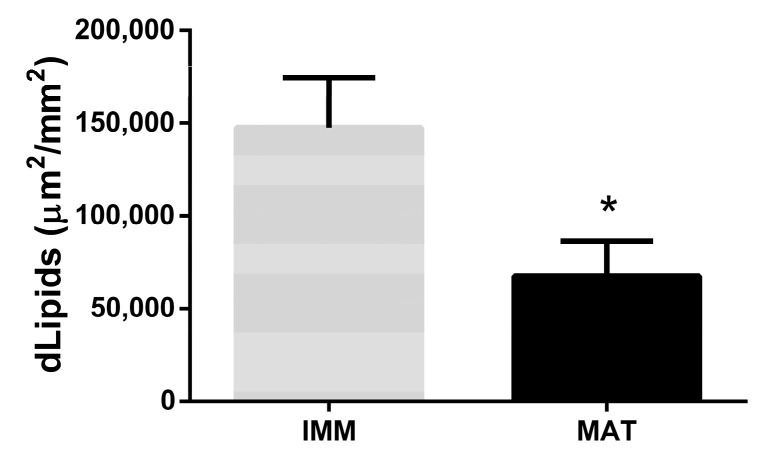
Difference in lipid density between livers of immature (*n* = 22) and mature (*n* = 28) females. Asterisk (*) indicates a significant statistical difference for *p*-value < 0.05.

**Table 1 animals-13-00269-t001:** Pearson correlations between MMCs and MMs density (μm^2^/mm^2^) and number per mm^2^ and fish biological parameters (LJFL and TW) (*n* = 50). Significant correlations (*p* < 0.05) are displayed in bold.

	Length	Weight	dMMC	dMM	MMC/mm^2^	MM/mm^2^
**Length**	1	**0.982066**	**0.627270**	**−0.628303**	**0.820992**	**−0.546776**
**Weight**	**0.982066**	1	**0.642745**	**−0.532282**	**0.817944**	−0.436625
**dMMC**	**0.627270**	**0.642745**	1	−0.276178	**0.550470**	−0.244675
**dMM**	**−0.628303**	**−0.532282**	−0.276178	1	−0.442146	**0.954888**
**MMC/mm^2^**	**0.820992**	**0.817944**	**0.550470**	−0.442146	1	−0.347469
**MM/mm^2^**	**−0.546776**	−0.436625	−0.244675	**0.954888**	−0.347469	1

**Table 2 animals-13-00269-t002:** Pearson correlations between lipid density (μm^2^/mm^2^) and fish biological parameters (LJFL and TW) (*N* = 50). Significant correlations (*p* < 0.05) are displayed in bold.

	Length	Weight	dLipids
**Length**	1	**0.982066**	**−0.52939**
**Weight**	**0.982066**	1	−0.40834
**dLipids**	**−0.52939**	−0.40834	1

## Data Availability

The data presented in this study are available on request from the corresponding author.

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
