# Peer review of "A Window of Vulnerability: Chronic Environmental Stress Does Not Impair Reproduction in the Swordfish Xiphias gladius"

_animals, 2023, doi:10.3390/ani13020269_

Round 1

Reviewer 1 Report

The study a window of vulnerability: chronic environmental stress does  not impair reproduction in the swordfish Xiphias gladius is nicely conducted. The experimental design is appropriate and the manucript is well written. The manucript can be accepted for pubication. The figures have poor resulution and should be fixed.

Author Response

Replies to Reviewer 1:

The study a window of vulnerability: chronic environmental stress does not impair reproduction in the swordfish Xiphias gladius is nicely conducted. The experimental design is appropriate and the manucript is well written. The manucript can be accepted for pubication. The figures have poor resulution and should be fixed.

Authors’ reply: we thank the Reviewer for the comments. As suggested, we improved the resolution of figures.

Reviewer 2 Report

The authors investigated the effect of chronic environmental stress on reproduction in the swordfish Xiphias gladius. The fundamental problem in the experiment design stopped me from reviewing this MS.

However, I have touched on some more points that can contribute to the improvement of this MS.

Minor comments

·       Line 14-15, please revise it.

·       Line 15-16, is against the points you made earlier. Please review this part and revise it.

·       Line 21, combining is not probably is a good option.

·       Line 22, downloaded? You mean you did not analyse yourself?> I read the MS and I found that you downloaded it. The variations in individuals are huge in omics data and it is impossible that you connect them to other histology data. Therefore, all data is not reliable and impossible to connect these data. Further, their samples are not enough at all to make any solution.

Kind regards

Author Response

The authors investigated the effect of chronic environmental stress on reproduction in the swordfish Xiphias gladius. The fundamental problem in the experiment design stopped me from reviewing this MS.

However, I have touched on some more points that can contribute to the improvement of this MS.

Authors’ reply: we thank the Reviewer for the comments. Below the point by point discussion

Minor comments

  • Line 14-15, please revise it.

Authors’ reply: The sentence was modified in a clearer form (Line 13-17)

  • Line 15-16, is against the points you made earlier. Please review this part and revise it.

Authors’ reply: The sentence was reviewed and modified (Line 17-20)

  • Line 21, combining is not probably is a good option.

Authors’ reply: The word “combining” was replaced with “coupling” (Line 25)

  • Line 22, downloaded? You mean you did not analyse yourself?> I read the MS and I found that you downloaded it. The variations in individuals are huge in omics data and it is impossible that you connect them to other histology data. Therefore, all data is not reliable and impossible to connect these data. Further, their samples are not enough at all to make any solution.

Authors’ reply: as stated in the M&M section we downloaded row data already deposited in a web site SwordfishOmics (http://www.swordfishomics.com) of our property. These raw data were used in a previous work performed by the same group, but in a different way. The present work intends to analyse these data using different bioinformatic tools, and such new analyses let us to evidence new differences between mature and immature females.

We improved introduction section adding new sentences explaining the importance of coupling transcriptomic and histological analyses and we referred to several papers in which these two approaches were used combined. In this light, we are still confident that is possible to connect these data.

Reviewer 3 Report

In general, the study is well described and the results are well shown. Just a few suggestions to try to improve the manuscript, and minor mistakes found through the text are listed down:

Abstract(s)

L13, 81 …swordfish females liver

L14, 32 most of energy > a lot of energy?

L17 Delete “also”

L32 Underscoring? Alternative word?

Introduction

L39 Delete comma

L45 …and responses to…

L76 Delete space before “(MMs)”

L91, 92 Separate numbers and units

L94, 122, 123, 124, 126 Use superscript format

L95, 96 Write compounds for males properly (use subscripts)

Results

L136 …and stress response…

L145, 147, 5 > Five

L147, 150, 154, 155 …are involved…

L151 8 > Eight

L154 3 > Three

L165, 171 bacteria?

L166 Send to a second paragraph?

L167 28 > Twenty-eight

Section 3.1.2. Decide if you write the biological processes using capitals or lower case letters.

L193, 194 (3x) Write linked > symbol and number

L199 …up regulated, while…

L201 Fig/Figure?

L220 …et al. (2020)…

L221 Among mature ones, …

L232 Use italics to write species name. Give here full meaning of H&E.

L239, 241, 244, 246, 247 (3x), 256, 257, 261, 271 (2x). Use superscript format

L266, 267, 270, 272, 275. Delete space before > symbol.

Fig 1. It seems too small. Try to include a bigger version. Write Expression in the axes.

Fig 3. Try to make the figure bigger using all the page width. Use of colours would be welcome.

Fig 4 and 5. Make graphs bigger (using the whole page) to make readable the axes text.

Table 1. Use superscript (4x)

Discussion

L286 …et al. (2014)…

L301 …et al. (2011)…

L340 significantly correlated on a negative way…?

References

Multiple mistakes are found. Check journal´s rule about.

Author Response

Replies to Reviewer 3:

In general, the study is well described and the results are well shown. Just a few suggestions to try to improve the manuscript, and minor mistakes found through the text are listed down:

Authors’ reply: we thank the Reviewer for the comments. Below the point-by-point discussion

Abstract(s)

 L13, 81 …swordfish females liver

Authors’ reply: The sentence was modified as suggested by the reviewer (Line 13 and 93)

L14, 32 most of energy > a lot of energy?

Authors’ reply: The sentence was modified as suggested by the reviewer (line 15 and 37)

L17 Delete “also”

Authors’ reply: The word “also” was deleted as suggested by the reviewer (Line 19-20)

L32 Underscoring? Alternative word?

Authors’ reply: The word “Underscoring” was changed with “highlighting” as suggested by the reviewer (Line 36)

Introduction

L39 Delete comma

Authors’ reply: comma was deleted as suggested by the reviewer (Line 43)

L45 …and responses to…

Authors’ reply: The sentence was modified as suggested by the reviewer (Line 49)

L76 Delete space before “(MMs)”

Authors’ reply: The space was deleted as suggested by the reviewer (Line 81)

L91, 92 Separate numbers and units

Authors’ reply: The numbers and units were separated as suggested by the reviewer (Line 103-104)

L94, 122, 123, 124, 126 Use superscript format

Authors’ reply: superscript format was used as suggested by the reviewer (line 106-108, 155-157)

L95, 96 Write compounds for males properly (use subscripts)

Authors’ reply: The sentence was modified as suggested by the reviewer (Line 107-108)

Results

L136 …and stress response…

Authors’ reply: The sentence was modified as suggested by the reviewer (Line 172)

L145, 147, 5 > Five

Authors’ reply: The sentence was modified as suggested by the reviewer (Line 178-191)

L147, 150, 154, 155 …are involved…

Authors’ reply: Sentences were modified as suggested by the reviewer (Line 178-191)

L151 8 > Eight

Authors’ reply: The sentence was modified as suggested by the reviewer (Line 178-191)

L154 3 > Three

Authors’ reply: The sentence was modified as suggested by the reviewer (Line 178-191)

L165, 171 bacteria?

Authors’ reply: “response to bacterium” is the official name of the Biological Process of Gene Ontology Analysis

L166 Send to a second paragraph?

Authors’ reply: The sentence was modified as suggested by the reviewer (Line 209)

L167 28 > Twenty-eight

Authors’ reply: The sentence was modified as suggested by the reviewer (Line 211)

Section 3.1.2. Decide if you write the biological processes using capitals or lower case letters.

Authors’ reply: we used capitals letters for each Biological processes (line 211-220)

L193, 194 (3x) Write linked > symbol and number

Authors’ reply: The sentence was modified as suggested by the reviewer (Line 238-242)

L199 …up regulated, while…

Authors’ reply: The sentence was modified as suggested by the reviewer (Line 244)

L201 Fig/Figure?

Authors’ reply: The sentence was modified as suggested by the reviewer (Line 266)

L220 …et al. (2020)…

Authors’ reply: The sentence was modified as suggested by the reviewer (Line 267)

L221 Among mature ones, …          

Authors’ reply: The sentence was modified as suggested by the reviewer (Line 261)

L232 Use italics to write species name. Give here full meaning of H&E.

Authors’ reply: The sentence was modified as suggested by the reviewer (Line 279-280)

L239, 241, 244, 246, 247 (3x), 256, 257, 261, 271 (2x). Use superscript format

Authors’ reply: The sentence was modified as suggested by the reviewer

L266, 267, 270, 272, 275. Delete space before > symbol.

Authors’ reply: space was deleted as suggested by the reviewer

Fig 1. It seems too small. Try to include a bigger version. Write Expression in the axes.

Authors’ reply: A new version of Figure 1 was included as suggested by the reviewer

Fig 3. Try to make the figure bigger using all the page width. Use of colours would be welcome.

Authors’ reply: the figure was improved, and colours were used as suggested by the reviewer

Fig 4 and 5. Make graphs bigger (using the whole page) to make readable the axes text.

Authors’ reply: bigger graphs were included as suggested by the reviewer

Table 1. Use superscript (4x)

Authors’ reply: The sentence was modified as suggested by the reviewer

Discussion

L286 …et al. (2014)…

Authors’ reply: The sentence was modified as suggested by the reviewer (Line 341)

L301 …et al. (2011)…

Authors’ reply: The sentence was modified as suggested by the reviewer (Line 356)

L340 significantly correlated on a negative way…?

Authors’ reply: The sentence was modified as suggested by the reviewer (Line 396-397)

References

Multiple mistakes are found. Check journal´s rule about.

Authors’ reply: references list was modified as suggested by the reviewer

Reviewer 4 Report

Please refer to your email regarding the paper review titled "A window of vulnerability: chronic environmental stress does not impair reproduction in the swordfish Xiphias gladius. The manuscript described the interaction of metabolism, stress response, immune system, and reproduction in immature and adult females. The experimental design was thorough and accurate, and the results were detailed and sturdy; nonetheless, histological figures should be adjusted or replaced with clearer ones. Personally, I believe it should be published after a single question. However, in order for the manuscript to flow well, it must be proofread by an expert or native English speaker.

It's a good idea to categorise and identify DEGs using bioinformatic tools. Can the authors, however, confirm the expression of the top five enriched DEGs using RT-qPCR?

Author Response

Replies to Reviewer 4:

Please refer to your email regarding the paper review titled "A window of vulnerability: chronic environmental stress does not impair reproduction in the swordfish Xiphias gladius. The manuscript described the interaction of metabolism, stress response, immune system, and reproduction in immature and adult females. The experimental design was thorough and accurate, and the results were detailed and sturdy; nonetheless, histological figures should be adjusted or replaced with clearer ones. Personally, I believe it should be published after a single question. However, in order for the manuscript to flow well, it must be proofread by an expert or native English speaker.

It's a good idea to categorise and identify DEGs using bioinformatic tools. Can the authors, however, confirm the expression of the top five enriched DEGs using RT-qPCR?

Authors’ reply: we thank the Reviewer for the comments. The manuscript has been submitted to a mother tongue colleague to improve the text fluency.

Histological figures were replaced and we also performed qPCR on five enriched DEGs as requested by the Reviewer and results were added to the manuscript.

Round 2

Reviewer 2 Report

Unofretuanlty, I saw the MS and I still believe that can not process further. Regretfully, I go with my previous decision.
Good luck with your future research
Kind regards

Author Response

RESPONSE TO REVIEWER 2

Unofretuanlty, I saw the MS and I still believe that can not process further. Regretfully, I go with my previous decision.
Good luck with your future research
Kind regards

Authors response: We apologize for reviewer 2's disappointment, but we still strongly believe in the possibility of combining histological and molecular studies and this point has been extensively illustrated in the manuscript.